# In Vivo Deposition of High-Flow Nasal Aerosols Using Breath-Enhanced Nebulization

**DOI:** 10.3390/pharmaceutics16020182

**Published:** 2024-01-28

**Authors:** Jeyanthan Jayakumaran, Gerald C. Smaldone

**Affiliations:** Division of Pulmonary, Critical Care and Sleep Medicine, Department of Medicine, Stony Brook University Medical Center, Stony Brook, NY 11794, USA; gerald.smaldone@stonybrookmedicine.edu

**Keywords:** breath-enhanced jet nebulizer, continuous nebulization, aerosol deposition, gamma scintigraphy

## Abstract

Aerosol delivery using conventional nebulizers with fixed maximal output rates is limited and unpredictable under high-flow conditions. This study measured regulated aerosol delivery to the lungs of normal volunteers using a nebulizer designed to overcome the limitations of HFNC therapy (*i-AIRE* (InspiRx, Inc., Somerset, NJ, USA)). This breath-enhanced jet nebulizer, in series with the high-flow catheter, utilizes the high flow to increase aerosol output beyond those of conventional devices. Nine normal subjects breathing tidally via the nose received humidified air at 60 L/min. The nebulizer was connected to the HFNC system upstream to the humidifier and received radio-labeled saline as a marker for drug delivery (^99m^Tc DTPA) infused by a syringe pump (mCi/min). The dose to the subject was regulated at 12, 20 and 50 mL/h. Rates of aerosol deposition in the lungs (µCi/min) were measured via a gamma camera for each infusion rate and converted to µg NaCl/min. The deposition rate, as expressed as µg of NaCl/min, was closely related to the infusion rate: 7.84 ± 3.2 at 12 mL/h, 43.0 ± 12 at 20 mL/h and 136 ± 45 at 50 mL/h. The deposition efficiency ranged from 0.44 to 1.82% of infused saline, with 6% deposited in the nose. A regional analysis indicated peripheral deposition of aerosol (central/peripheral ratio 0.99 ± 0.27). The data were independent of breathing frequency. Breath-enhanced nebulization via HFNC reliably delivered aerosol to the lungs at the highest nasal airflows. The rate of delivery was controlled simply by regulating the infusion rate, indicating that lung deposition in the critically ill can be titrated clinically at the bedside.

## 1. Introduction

Patients with hypoxic respiratory failure are often treated with oxygen using a high-flow nasal cannula (HFNC). In addition to providing high FiO_2_, reduced dead space and positive airway pressure, the nasal cannula offers a pathway to the respiratory tract that may allow targeted therapy to the lungs with aerosolized drugs, such as bronchodilators or vasodilators. While the HFNC route affords the opportunity to treat critically ill patients before intubation, delivery of aerosols via this route is uncertain, and studies using modern conventional nebulizers report decreased aerosol delivery at higher flows. In 2017, Dugernier and colleagues quantified aerosol delivery and deposition in normal subjects receiving high-flow aerosols using jet and vibrating mesh devices. They determined the distribution of aerosol losses and lung deposition following a treatment with a single nebulizer charge of 4 mL Diethylenetriamine pentaacetate (^99m^Tc-DTPA) at 30 L/min gas flow. Deposition in the lungs ranged from 0.7 to 4.4%, indicating that most of the generated aerosol was lost, reflecting the difficulty involved in moving particles through clinical high-flow systems [1]. With bolus aerosol delivery (e.g., filling the nebulizer with a drug solution), the delivered dose is uncertain because of the many factors affecting delivery, particularly the losses in high-flow cannula circuits. If the drug is infused continuously, there is the potential to control therapy because, as long as the nebulizer maximal output is not exceeded, whatever is infused is nebulized [2]. Model studies predict greater losses at higher gas flows. Moon et al., in an anatomically correct head model, found that delivery to the lungs progressively decreased from 10 to 60 L/min [3]. This form of aerosol therapy uses drugs off label following local hospital protocols. In the critically ill, rapidly acting intravenous drugs are often given via continuous infusion and titrated in a dose response manner. Therefore, therapy is individualized. Can this approach work for inhaled therapy? Conventional nebulizers can support continuous nebulization, but the range of therapy for a given concentration of drug is limited by limitations in nebulizer output (e.g., 10 mL/h). For conventional mesh nebulizers, some authors suggest that nasal flows be reduced facilitating aerosol delivery [4,5].

Breath-enhanced nebulization incorporates the high gas flow moving to the patient internally in the nebulizer chimney, using the energy of the high-flow gas to increase the nebulizer output beyond those of conventional devices. Directing the high flow first to the nebulizer, before delivery to the patient, creates a Venturi effect in the nebulizer chimney. The Venturi effect lowers pressure in the nebulizer, increasing the rate of fluid transport from the nebulizer bowl to the region of the nebulizer anvil, resulting in a gas-flow-dependent increase in aerosol production [6]. Moon et al. tested this hypothesis in a bench study of HFNC therapy. That study proved that breath enhancement, under high-flow conditions, markedly increased aerosol production beyond that seen from a conventional mesh nebulizer (approx. 5 times the conventional output). In addition, they found that aerosol delivery far exceeded that of mesh technology, and, of equal importance, delivery was regulated via the infusion rate from an external regulated source of drug, such as a syringe pump or regulated IV bag [3]. Under the most difficult clinical conditions (60 L/min catheter gas flow) used in the sickest patients, the breath-enhanced device may allow titratable aerosol therapy at the bedside, with doses ranging for more than an order of magnitude. The present paper was designed to confirm the observations of Moon et al. in vivo, testing the potential of breath enhancement in humans. The chosen device (an *i-AIRE*, InspiRx, Inc., Somerset, NJ, USA) was a prototype breath-enhanced jet nebulizer developed for hospital use. This nebulizer is a candidate for United States Food and Drug Administration 510(k) clearance for marketing and is not yet commercially available. The protocol measured the serial deposition of radio-labeled test aerosols in the lungs of normal volunteers. The delivery of the aerosol was controlled using a syringe pump, with the goal being to deliver drug to the lung over a wide range of precisely controlled infusion rates.

## 2. Materials and Methods

### 2.1. Experimental Design

Nine healthy adult volunteers were enrolled in this study after signing an IRB-approved informed consent form. After baseline spirometry, subjects were seated in front of a gamma camera (Maxi Camera 400; General Electric, Horsholm, Denmark; Model 604/150/D; Power Computing, Austin TX; Nuclear Power, version 3.0.7; Scientific Imaging, Inc., Thousand Oaks, CA, USA) and a 15 min background image obtained with the camera set for 99mTechnetium (^99m^Tc). Following background imaging, subjects held a double-walled rectangular Lucite container with a space between the walls, filled with approx. 5mCi of radioactivity diffused throughout, and a two-minute transmission image was obtained. The transmission image was utilized to aid in determinig the optimal positioning of the subject and estimate regional lung volume for use in regional lung deposition calculations [7,8].

The experimental setup is outlined in Figure 1.

Normal saline was used as a marker of a test drug. Two test solutions were prepared. Non-radioactive saline in a 1 L IV bag was hung on one side of an Alaris infusion pump (Alaris Pump Module, Becton, Dickinson and Co., Franklin Lakes, NJ, USA). On the other side, equipped with a syringe infusion pump, radio-labeled normal saline was mixed with 99mTechnetium bound to Diethylenetriamine pentaacetate (^99m^Tc-DTPA) to create a radioactive solution with approx. 3.5 mCi/mL. High concentrations of ^99m^Tc-DTPA were used to overcome anticipated losses in the HFNC circuit.

Volunteers were outfitted with a high-flow nasal cannula (Optiflow, Fisher and Paykel, Auckland, New Zealand) and exposed to a test airflow of 20 L/min that was gradually increased to 60 L/min over 5–10 min. In addition to the test air flow, a saline infusion using the IV bag introduced non-radioactive saline into the nebulizer (*i-AIRE* (InspiRx, Inc. Somerset, NJ, USA, activated with 5 L/min air at 50 PSIG). The saline was infused at increasing rates from 20 to 50 mL/h over 10 min. Volunteers were instructed to breathe normally via the nose. Once comfortable on a gas flow of 60 L/min and infusion rate of 50 mL/h, the infusion was switched to the syringe side of the pump, and a radio-labeled continuous infusion of saline was initiated at 12 mL/h, a rate suggested by Moon et al. that would provide a low rate of aerosol delivery to the airway. The number of tidal breaths was counted over the duration of the radio-labeled infusions (frequency of breathing (breaths/min)). Subjects were monitored using the gamma camera, and once the count rate of the image was sufficient for accurate scanning, the infusion was stopped, and a five-minute static image taken. With image acquisition complete, the infusion protocol with imaging was repeated at 20 mL/h and 50 mL/h. Each run from the start of infusion to the completion of the static scan took approximately 10 min. After the completion of the last lung image, the subject changed position, and a lateral scan of the head was performed to measure cumulative nasal deposition.

### 2.2. Analysis

Deposited radioactivity was quantified by normalizing counts for each image to one minute. Activity from both lungs was included in all lung deposition calculations because most upper airway activity remained in the nose and was not swallowed. When analyzing the regional distribution of deposited activity, one lung was excluded in 3 subjects because of esophageal or stomach activity that overlapped the central airways. After correction for room background on the first deposition image, succeeding deposition images were corrected via the subtraction of the activity from the preceding image and decay corrected. Measured activity was converted to drug activity by using one minute as the basis for calculation. Deposition counts from gamma camera images were corrected for chest wall attenuation using Equation (1), as described by Fleming et al. [9,10]
AF_gm_ = 0.0562 BMI + 0.907 (1)
where AF_gm_ represents attenuation factor for the geometric mean, and BMI is body mass index in kg/m^2^. The deposition rate expressed in µg NaCl/min for each infusion rate was calculated from Equation (2), using the salt content of normal saline (9000 µg/L). The first term of Equation (2) represents the µCi of radioactivity deposited per minute. The second term converts the deposited radioactivity to µg of NaCl, which served as the surrogate drug.
(2)$$DR=\frac{A*AF}{E*t\;}/\frac{SC}{TV*9000}$$
*DR* is the deposition rate in µg NaCl/min, *A* is activity (counts) deposited normalized to one minute (background and decay corrected) *AF*, the Fleming correction factor corrects the counts for chest wall attenuation, *E* is camera efficiency (counts/min/µCi, measured with activity placed on camera face), *t* is total time of the infusion (min), *SC* is syringe charge (µCi) and *TV* is total volume of solution (mL). Camera efficiency was measured using a Pari filter (Pari, Sternberg, Germany) infused with a known amount of ^99m^Tc-DTPA each experimental day. The DR represents the lung dose (e.g., drug deposited/min). With the lung dose calculated, the efficiency of drug delivery (DE) for the high-flow nasal cannula system could be calculated by Equation (3)
DE% = [DR (µCi/min)/mCi infused] ∗ 100(3)

The DE represents the percentage of drug deposited in the lung compared to the total infused. Nasogastric deposition efficiency, consisting of cumulative nasal and stomach deposition for the entire experiment, was estimated in a similar fashion. During the lateral imaging of the head, the subject’s nose was essentially on the camera face, and deposited activity was measured as counts/E. For small amounts of stomach activity, radioactivity attenuation correction across individuals was estimated to be double that of the lung (2AF) [7].

Regional deposition was quantified using central to peripheral ratios (C/P), as outlined in detail by Smaldone et al. [8] and Samuel et al. [7]. As shown in the sketch in Figure 1, whole lung and central regions were hand drawn around the final deposition image and superimposed onto the transmission image. The central region of interest encompassed approx. 1/3 of the lung containing the central airways. The ratio of counts in the central to peripheral region for the deposited aerosol particles (aC/P) was normalized using the transmission image ratio (tC/P) to correct for regional lung volume, resulting in the ratio of deposited particles per unit of lung volume (sC/P). Using this technique, sC/P of 1.0 represents particles deposited in small airways and alveoli [7,8]. 

The statistical significance between groups was determined using the Wilcoxon test (GraphPad Prism for Mac OS X (GraphPad Software version 10.1.1, San Diego, CA, USA)). The results are reported as a mean ± standard deviation. 

## 3. Results

The baseline lung function and anthropomorphic details of the test subjects are listed in Table 1.

An example of serial deposition images for a test subject at the three test infusion rates is shown in Figure 2. Activity on the 5 min static subtraction images increases sequentially with 69.7 µCi, 142.6 µCi and 348.9 µCi deposited for the 12 mL/h, 20 mL/h and 50 mL/h infusions, respectively. In general, the lungs were well defined with minimal stomach activity unless some nasal activity was swallowed towards the end of the test protocol. The images qualitatively suggest peripheral deposition with no activity seen in central airways. The lateral nasal image captured at the end of the aerosol inhalations is also shown. The pixel sensitivity was changed to visualize movement during this study, which took about 30 min for the three infusions. Some radioactivity moved deeper into the nasal pharynx, but no significant activity entered the esophagus or stomach. This behavior was typical in most patients.

Figure 3 describes the individual deposition rates, expressed as µg NaCl/min at each infusion rate plotted on a log scale to bring out the details. The mean deposition rates (7.84 ± 3.2, 43.0 ± 12 and 136 ± 45 µg NaCl/min) significantly increased with infusion rates, e.g., 12 to 20 mL/h (*p* = 0.0039) and 20 to 50 mL/h (*p* = 0.0039).

Table 2 displays the deposition efficiency (%) of the lungs for each infusion rate and combined nasogastric efficiency measured at the end of the experiment. Deposition efficiency significantly increased between the 12 mL/h efficiency and the efficiencies measured at the higher infusion rates, e.g., 0.44 ± 0.2% vs. 1.45 ± 0.40% at 20 mL/h or 1.82 ± 0.61% at 50 mL/h (*p* = 0.0039). Differences between deposition efficiency at 20 and 50 mL/h were not significant (*p* = 0.0742). On average, 6.23 ± 1.6% of total infused radioactivity from the syringe pump was deposited in the nasogastric region (5.95 ± 1.6%nasal/0.28 ± 0.5% stomach).

Regional lung deposition (Table 3), represented by C/P ratios normalized for volume (sC/P ratio), averaged 0.99 ± 0.27 across both lungs. Figure 4 illustrates the relationship between the deposition rate (in µg NaCl/min) and respiratory frequency at the various infusion rates. While the effects of changes in infusion rate are apparent, there was no correlation between the rate of deposition and breathing frequency; *p* = *0*.885 and R^2^ = 0.0032 at 12 mL/h, *p* = 0.753 and R^2^ = 0.015 at 20 mL/h and *p* = 0.884 and R^2^ = 0.0033 at 50 mL/h.

## 4. Discussion

This study demonstrates that HFNC therapy coupled to breath-enhanced nebulization can target the lungs with aerosol at the highest gas flow used clinically. Delivery to the peripheral airways can be controlled over a wide range of deposition rates defined by the rate of infusion. As shown in Figure 3, rates of drug delivery ranged over an order of magnitude, demonstrating the potential for individualized therapy. Deposition rates per se reflect the dose to the lungs per minute. For fast-acting drugs with short half-lives (e.g., epoprostenol), clinical responses could be assessed after reaching a steady state for each infusion rate and adjusted up or down as needed. For drugs with longer half-lives such as antibiotics, clinically relevant lung doses could be delivered using an appropriate mean delivery rate for a fixed number of minutes (e.g., dose rate (t) = total lung dose). 

Our data in Figure 4 suggest that during HFNC therapy, the physiology of particle transport to the terminal airways is different than that during tidal breathing. While there were some differences in deposition rate from patient to patient, the variability is much less than that reported for typical aerosol delivery of small particles for tidal breathing, where differences in breathing frequency can result in significant changes in deposition. Bennett et al. found that for small particles, breath-by-breath deposition in normal subjects varied as 1/f^2^, where f was defined as breathing frequency (breaths/min) [11]. During HFNC therapy, contrary to observations in spontaneously breathing normal subjects, deposition was not sensitive to breathing frequency. Fine particles inhaled during tidal breathing were deposited primarily via gravitational settling, governed by particle residence time in the airways, a function of breathing frequency. In the study of Moon et al., the mass median aerodynamic diameter of particles leaving the nasal catheter were measured as 1.10 µm ± 0.03, which, during tidal breathing, should be sensitive to breathing frequency [3]. Deposition during HFNC therapy may be governed by different mechanisms. With HFNC therapy, convective air flow effectively reduces dead space, and particles may be transported to the peripheral lung via convection created by the high-flow gases, which are full of aerosol particles, rather than via a tidal bolus of aerosol that needs to penetrate through the dead space. Deposition in the distal lung may be determined by an exchange between the aerosol front deep in the lung and alveolar gas, a process that may be less sensitive to tidal breathing. Consistent with this hypothesis, the C/P ratios (Table 3) in our subjects were indicative of deposition in the smallest airways and alveoli [8]. Further work is necessary to understand particle transport within the lungs during HFNC therapy to assess its potential for targeting small airways, particularly in disease. 

While lung deposition was controlled, the rate of drug delivery to the lungs was much less than the rate of delivery to the nose (Table 2). We did not record serial nasal delivery for each infusion rate, but for the total experiment, the deposition efficiency of the nose was approximately 10 times higher. In future studies testing active drugs, systemic drug effects may result from nasal deposition. However, previous studies using aerosol bolus delivery through the high-flow nasal cannula found significant changes in FEV_1_, implying that patients received therapeutic lung dosages, but importantly, adverse events such as serious tachycardia did not occur. Observed increases in the heart rate were not above those typically seen following oral bronchodilator therapy, e.g., five to eight beats per minute [12,13]. Theoretically, the nasal passages communicate with the brain through the olfactory mucosa [14], raising concerns that medications given through HFNC may cause neurologic effects. McPeck et al. [2] tested deposition using breath-enhanced nebulization in an anatomically correct head model and reported deposition primarily in the perinostril area (the region of squamous and respiratory mucosa), which does not encounter the blood–brain barrier. In our study, nasal deposition tended to remain fixed in location (as in Figure 2, where minimal clearance from the nose was observed during a 10–30 min experiment). Ultimately, deposited drug either drips out of the nose anteriorly or particles are eventually swallowed due to the ciliated lining of the nasal passageway, moving particles away from the olfactory epithelium, a mucociliary clearance process like that seen in the lower respiratory tract [15].

Our protocol tested HFNC delivery at the highest gas flow used for critically ill patients, e.g., 60 L/min of gas. However, these data apply to normal subjects, and the behavior of aerosols remains to be demonstrated in patients with lung disease. HFNC aerosol delivery is inefficient, with significant losses across the circuit, as well as at the nasal interface. Over 95% of the infused drug is lost, as illustrated by the deposition efficiencies reported in Table 2. Breath-enhanced nebulization overcomes these inefficiencies. Without breath-enhanced increases in nebulizer output with increases in infusion rate, the losses in the high-flow circuit, including nasal deposition, would be difficult to overcome. 

The present paper confirms the in vitro predictions of Moon et al. in vivo [3]. They examined the output of the breath-enhanced jet nebulizer compared to a vibrating mesh device across a wide array of infusion and gas flow rates. The findings in their model closely resemble those observed in the present in vivo study. For example, Moon et al. found more than an order of magnitude increase in the rate of aerosol delivery with increases in infusion rate between 12, 20 and 50 mL/h at 60 L/min of flow, similar to our findings in vivo (17-fold increase). Their bench model may apply to changes in the deposition rate for other combinations of gas flow and infusion flow rates.

Critically ill patients in the hospital are being placed on high-flow nasal cannula for the treatment of a variety of conditions that may be treated with continuous or bolus nebulization using conventional jet or mesh nebulizers. Reviews of in vitro and in vivo testing have concluded that the nasal gas flow rate must be reduced in order to mitigate aerosol losses, a maneuver with some complexity and risk [4,5]. It is true that aerosol losses increase with increases in high-flow rates. However, our findings show that breath-enhanced jet nebulization, combined with continuous infusion, facilitates therapy at the highest nasal flows. Patients with acute respiratory distress syndrome or other conditions requiring high levels of oxygen and/or flow support can be placed on a continuous infusion of pulmonary vasodilator, titrated to therapeutic effect while maintaining high flow. Similarly, patients with an obstructive lung disease in exacerbation may be treated with albuterol, while those with pulmonary infection can receive inhaled antibiotics, all through the life-supporting HFNC without needing to decrease gas flow. 

In conclusion, the present paper shows that during HFNC therapy, breath-enhanced nebulization can deliver controlled amounts of aerosol particles to the distal airways of human subjects. Future studies will determine if this form of aerosol delivery is more effective than conventional aerosol therapy.

## Figures and Tables

**Figure 1 pharmaceutics-16-00182-f001:**
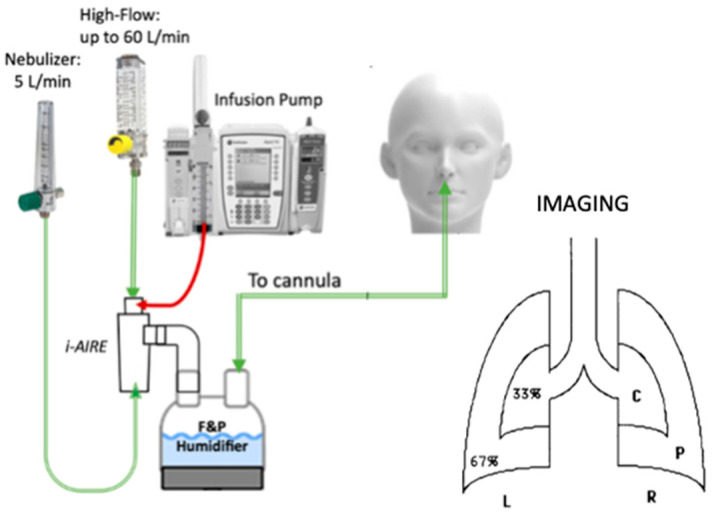
Experimental setup modified from Moon et al. [3], with the regional lung analysis outlined (C = central, P = peripheral, L, R = left and right lungs) [7,8].

**Figure 2 pharmaceutics-16-00182-f002:**
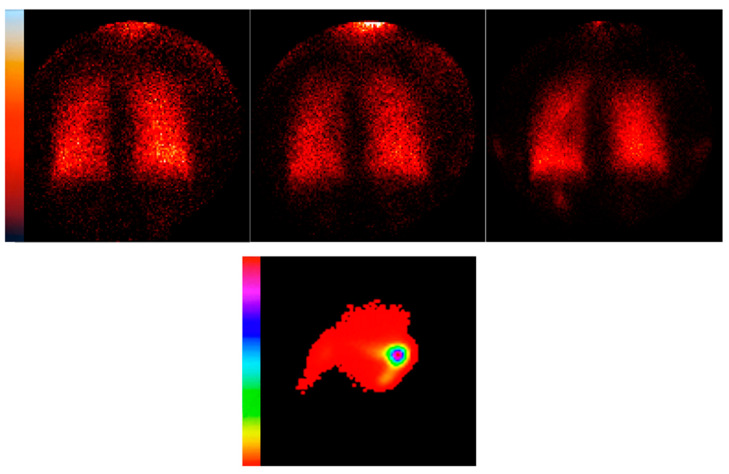
Serial 5 min deposition images created via sequential subtraction for subject 6, depicting, from left to right, 12 mL/h (69.7 µCi), 20 mL/h (142.6 µCi) and 50 mL/h (348.9 µCi) ((**upper**) panel). The lateral nasal scan obtained at the end of aerosol inhalation is shown: the subject is facing to the right ((**lower**) panel, camera spectrum changed to visualize activity distribution).

**Figure 3 pharmaceutics-16-00182-f003:**
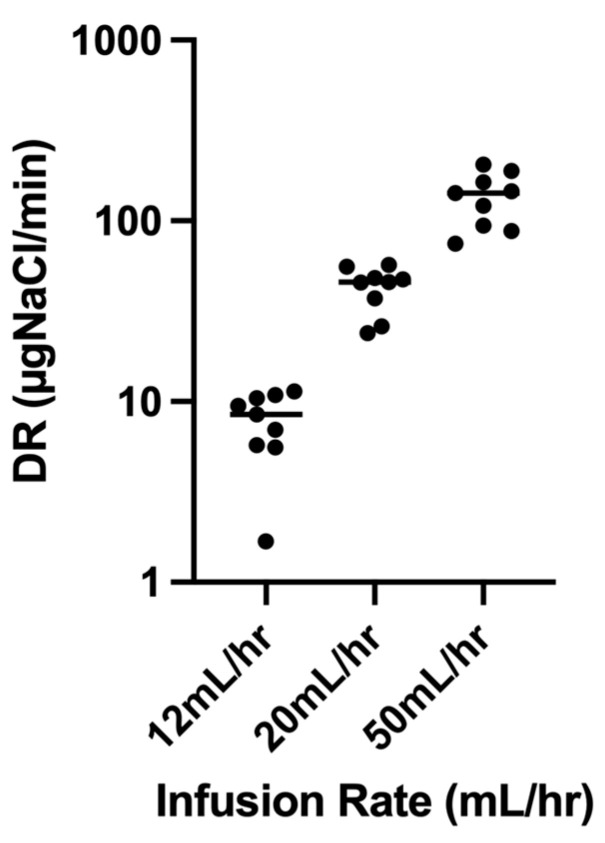
Log of Deposition rates (µgNaCl/min) as a scatter plot at the three designated infusion rates: (mean ± SD) 7.84 ± 3.2 at 12 mL/h, 43.0 ± 12 at 20 mL/h and 136 ± 45 at 50 mL/h.

**Figure 4 pharmaceutics-16-00182-f004:**
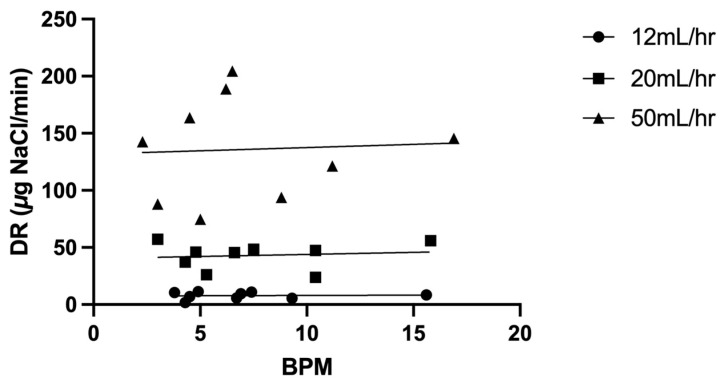
Deposition rate (µg NaCl/min) plotted against breathing frequency (BPM breaths per min) at each infusion rate with the line of best fit: 12 mL/h [closed circle], 20 mL/h [closed triangle] and 50 mL/h [closed square].

**Table 1 pharmaceutics-16-00182-t001:** Spirometry of volunteers % = % predicted.

Patient ID	Sex	Age	FEV_1_ (%)	FVC (%)	FEV_1_/FVC
1	M	75	115%	103%	0.81
2	F	74	84%	90%	0.70
3	F	59	92%	93%	0.77
4	F	36	123%	130%	0.78
5	M	60	101%	97%	0.81
6	F	31	100%	94%	0.89
7	M	61	86%	84%	0.80
8	M	73	98%	100%	0.71
9	F	72	112%	97%	0.87
Mean ± SD		60 ± 16	101 ± 13%	99 ± 13%	0.79 ± 0.06

**Table 2 pharmaceutics-16-00182-t002:** Deposition efficiency at each infusion rate and overall nasogastric efficiency.

Patient ID	12 mL/h (%)	20 mL/h (%)	50 mL/h (%)	Nasogastric (%)
1	0.33	1.55	1.60	6.75
2	0.09	0.87	1.01	5.58
3	0.39	1.27	1.94	5.67
4	0.61	1.95	1.15	5.08
5	0.53	1.50	2.57	6.75
6	0.63	1.57	2.14	4.38
7	0.49	1.88	1.96	8.76
8	0.31	0.82	1.25	8.50
9	0.60	1.65	2.73	4.61
Mean ± SD	0.44 ± 0.2	1.45 ± 0.40	1.82 ± 0.61	6.23 ± 1.6

**Table 3 pharmaceutics-16-00182-t003:** Central to peripheral ratios: aerosol (aC/P), transmission image (tC/P) and normalized (sC/P).

Subject ID	RaC/P	RtC/P	RsC/P	LaC/P	LtC/P	LsC/P	Total sC/P
1	0.27	0.41	0.66	0.31	0.22	1.43	1.04
2	0.47	0.55	0.86	0.50	0.57	0.89	0.87
3	0.42	0.39	1.07	0.63	0.50	1.25	1.16
4 *	0.49	0.66	0.74	nd	nd	nd	0.74
5	0.36	0.34	1.04	0.55	0.49	1.12	1.08
6	0.31	0.33	0.94	0.44	0.28	1.58	1.26
7 *	0.45	0.56	0.80	nd	nd	nd	0.80
8 *	nd	nd	nd	0.42	0.76	0.56	0.56
9	0.53	0.39	1.35	0.70	0.50	1.40	1.38
Mean ± SD	0.41 ± 0.09	0.45 ± 0.12	0.93 ± 0.22	0.51 ± 0.13	0.47 ± 0.18	1.18 ± 0.35	0.99 ± 0.27

R = right lung; L = left lung; C/P = ratio of counts in central (C) and peripheral (P) regions; a = aerosol, t = transmission; s = a/t; * one lung not included in regional analysis (nd = not done) due to stomach or esophageal activity.

## Data Availability

The data presented in this study are available in the main body of the text.

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
