# Peer review of "In Vivo Deposition of High-Flow Nasal Aerosols Using Breath-Enhanced Nebulization"

_pharmaceutics, 2024, doi:10.3390/pharmaceutics16020182_

Round 1

Reviewer 1 Report

Comments and Suggestions for Authors

May be published in Pharmaceutics after major revision

In their manuscript entitled “In Vivo Deposition of High Flow Nasal Aerosols Using Breath Enhanced Nebulization”, Jeyanthan Jayakumaran and Gerald C Smaldone report about HFNC therapy combined with breath-enhanced nebulization, which the Authors claim allow to deliver controlled amounts of aerosol particles to the distal airways of human subjects.

Appreciation

In my opinion, this study is worth of interest and fits well in the scope of the Journal Pharmaceutics. However, although the manuscript is concise and interesting to read, I think it should be revised, to improve its writing, provide more explanation, as well as refer to more previously published relevant studies, which should increase its impact. Accordingly, I recommend a major revision of this manuscript, to address the abovementioned remarks that are more detailed with the specific comments listed below.

Specific comments

The writing can be enhanced in different ways, with respect to the style (some sentences should be rephrased), typo errors (e.g. duplicated spaces, parenthesis), definition of several abbreviations, etc.

Author state at the end of the abstract “Breath-enhanced nebulization via HFNC reliably and predictably delivered a wide range of aerosolized surrogate medication to the lungs, determined simply by regulating the infusion rate”. This writing is confusing/erroneous and must be rephrased since the current study ‘only’ used radiolabeled saline as a marker for drug delivery, but not any therapeutic substrate that could make sense to be delivered using this method, as discussed by the authors later in the manuscript.

The Introduction may be developed by more referring to other earlier studies.

 “Conventional nebulizers” – as mentioned by the authors several times in their manuscript – should be specified.

A Figure or Scheme depicting the experimental set up would allow a better understanding how the study was conducted.

The discussion should be developed to better discuss about the limitations of the study. The rational for focusing on the non-conventional nebulizer used in this study can be criticized. The parallel evaluation of other ‘more conventional devices’ would have been insightful, as the benefits claimed by the Authors could have been better highlighted.

The Reference section is quite limited, including only 12 references to previously published papers. The Introduction section only contains 3 citations. More relevant publications should be added to better support the authors claims.

Minor comments

Please check/revise/add:

- Color scale in Figure 1

- Y axis in Figure 2

- BPM in Figure 3

- Not determined data in Table 3 should be mentioned “nd”

- Nasogastric pictures may be added.

- More pictures of subjects may be provided (as SI).

Comments on the Quality of English Language

Moderate editing of English language required

Author Response

The writing can be enhanced in different ways, with respect to the style (some sentences should be rephrased), typo errors (e.g. duplicated spaces, parenthesis), definition of several abbreviations, etc.

Author state at the end of the abstract “Breath-enhanced nebulization via HFNC reliably and predictably delivered a wide range of aerosolized surrogate medication to the lungs, determined simply by regulating the infusion rate”. This writing is confusing/erroneous and must be rephrased since the current study ‘only’ used radiolabeled saline as a marker for drug delivery, but not any therapeutic substrate that could make sense to be delivered using this method, as discussed by the authors later in the manuscript.

Rephrased as suggested.

The Introduction may be developed by more referring to other earlier studies.

We have rewritten the Introduction adding more references

 “Conventional nebulizers” – as mentioned by the authors several times in their manuscript – should be specified. 

Comparisons have been performed in previous studies particularly Moon et al. where our group compared breath enhanced nebulizer to vibrating mesh.  Experiments described in expanded introduction and discussion of referenced papers.

A Figure or Scheme depicting the experimental set up would allow a better understanding how the study was conducted. 

Figure 1 added.

The discussion should be developed to better discuss about the limitations of the study. The rational for focusing on the non-conventional nebulizer used in this study can be criticized. The parallel evaluation of other ‘more conventional devices’ would have been insightful, as the benefits claimed by the Authors could have been better highlighted.

We agree with the Reviewer that the value of the prototype device can be better demonstrated by comparing it to conventional systems.  Before attempting this in vivo study, our group performed an extensive in vitro study comparing the protype device to a conventional mesh nebulizer demonstrating the superior performance of the breath enhanced system.  The study by Moon et al proved that breath enhancement increased output at least 5 times that of mesh technology and, in an anatomically correct model of a human head delivered significant quantities of aerosol to the model under conditions where controlled delivery was not possible using the conventional device. The present study was designed to confirm the device’ function in vivo under the most demanding high flow condition (60L/min).  We have expanded our description of their work in the Introduction.

The Reference section is quite limited, including only 12 references to previously published papers. The Introduction section only contains 3 citations. More relevant publications should be added to better support the authors claims.

There are not many papers addressing this problem quantitatively, we have added several references.  Only a few studies have been performed with scintigraphy. 

Minor comments

Please check/revise/add:

- Color scale in Figure 1

Added

- Y axis in Figure 2

Y axis is defined on the Figure and in the text

- BPM in Figure 3

changed to breathing frequency (breaths/min)

- Not determined data in Table 3 should be mentioned “nd”

changed

- Nasogastric pictures may be added. 

 nasal image of patient shown in new Fig 2

Reviewer 2 Report

Comments and Suggestions for Authors

The paper presents an experimental study on using a breath-enhanced high-flow nebulizer to achieve better deposition of aerosol into lung. In general, the reported is interested to drug delivery research and aligns with the journal scope. However, there are some suggestions to improve its readability.

i) The originality should be articulated. What are the new findings comparing to Moon’s design?

ii) As a new technology, the principles for the improvement of the deposition should be introduced, at least briefly. Some schematics of the experimental equipment will be helpful to readers.

iii) The equations can be better formatted.

iv) The discussion part is too long, can be divided into several sections and a separate conclusion section.

V) References to the figures and tables should be added in some discussion.

Comments on the Quality of English Language

Minor eiditing.

Author Response

The paper presents an experimental study on using a breath-enhanced high-flow nebulizer to achieve better deposition of aerosol into lung. In general, the reported is interested to drug delivery research and aligns with the journal scope. However, there are some suggestions to improve its readability.

  1. i) The originality should be articulated. What are the new findings comparing to Moon’s design?

We have rewritten the Introduction to address this adding more references. 

We agree with the Reviewer that the value of the prototype device can be better demonstrated by comparing it to conventional systems.  Before attempting this in vivo study, our group performed an extensive in vitro study comparing the protype device to a conventional mesh nebulizer demonstrating the superior performance of the breath enhanced system.  The study by Moon et al proved that breath enhancement increased output at least 5 times that of mesh technology and, in an anatomically correct model of a human head delivered significant quantities of aerosol to the model under conditions where controlled delivery was not possible using the conventional device. The present study was designed to confirm the device’ function in vivo under the most demanding high flow condition (60L/min).  We have expanded our description of their work in the Introduction

  1. ii) As a new technology, the principles for the improvement of the deposition should be introduced, at least briefly. Some schematics of the experimental equipment will be helpful to readers.

 Introduction expanded.

iii) The equations can be better formatted.

  1. iv) The discussion part is too long, can be divided into several sections and a separate conclusion section.
  2. V) References to the figures and tables should be added in some discussion.

In response to these points, as outlined above we have rewritten the Introduction and Discussion, expanded the description of the equations with and added references.

Round 2

Reviewer 1 Report

Comments and Suggestions for Authors

The authors took into account my previous comments to provide an improved version of their original manuscript. Therefore, I think it is now suitable for publication in Pharmaceutics.

Comments on the Quality of English Language

Minor editing of English language required